# The Polycomb group protein CBX6 is an essential regulator of embryonic stem cell identity

Alexandra Santanach[1,2], Enrique Blanco [1,2], Hua Jiang[3], Kelly R. Molloy[4], Miriam Sansó[1,2], John LaCava [3,5], Lluis Morey[1,2,6] & Luciano Di Croce [1,2,7]

Polycomb group proteins (PcG) are transcriptional repressors that control cell identity and development. In mammals, five different CBX proteins associate with the core Polycomb repressive complex 1 (PRC1). In mouse embryonic stem cells (ESCs), CBX6 and CBX7 are the most highly expressed CBX family members. CBX7 has been recently characterized, but little is known regarding the function of CBX6. Here, we show that CBX6 is essential for ESC identity. Its depletion destabilizes the pluripotency network and triggers differentiation. Mechanistically, we find that CBX6 is physically and functionally associated to both canonical PRC1 (cPRC1) and non-canonical PRC1 (ncPRC1) complexes. Notably, in contrast to CBX7, CBX6 is recruited to chromatin independently of H3K27me3. Taken together, our findings reveal that CBX6 is an essential component of ESC biology that contributes to the structural and functional complexity of the PRC1 complex.

[1] Centre for Genomic Regulation (CRG), The Barcelona Institute of Science and Technology, Dr. Aiguader 88, Barcelona 08003, Spain. [2] Universitat Pompeu Fabra (UPF), Barcelona 08003, Spain. [3] Laboratory of Cellular and Structural Biology, The Rockefeller University, New York, NY 10065, USA. [4] Laboratory of Mass Spectrometry and Gaseous Ion Chemistry, The Rockefeller University, New York, NY 10065, USA. [5] Institute for Systems Genetics and Department of Biochemistry and Molecular Pharmacology, New York University School of Medicine, New York, NY 10016, USA. [6] Sylvester Comprehensive Cancer Center, Department of Human Genetics, University of Miami Miller School of Medicine, Miami, FL 33136, USA. [7] ICREA, Pg. Lluis Companys 23, Barcelona 08010, Spain. Correspondence and requests for materials should be addressed to L.D.C. (email: luciano.dicroce@crg.eu)

Polycomb group proteins (PcG) play essential roles in the regulation of gene expression during cell fate specification and embryonic development[1, 2]. PcG proteins associate in complexes that are classified in two major functional groups, the Polycomb repressive complex 1 (PRC1) and the Polycomb repressive complex 2 (PRC2), both of which are catalytically active[3, 4]. PRC2 contains the histone methyltransferase EZH1/2, which di- or tri-methylates lysine 27 of histone H3 (H3K27me2/3)[5], a mark associated with transcriptionally repressed chromatin, whereas PRC1 monoubiquitinates lysine 119 of histone H2A (H2AK119ub) through the E3 ligases RING1A/B[6].

The canonical mammalian PRC1 (cPRC1) contains four core subunits: the E3 ligase Ring1 Drosophila ortholog RING1A/B, one of the orthologs of the posterior sex combs (Psc) (PCGF1–6), a polyhomeotic (Ph) ortholog (PHC1-3), and a Polycomb (Pc) ortholog (CBX2/4/6-8)[3, 7]. However, six major groups of PRC1 complexes have been defined, according to the PCGF subunit associated with it[8]. The five so-called non-canonical PRC1s (ncPRC1) likewise have numerous additional protein subunits, such that each complex contains a RING1A/B subunit, a distinct PCGF subunit, and a unique set of associated polypeptides. Importantly, different PRC1 complexes have been found in multiple cellular contexts, suggesting distinct functional roles for the specific subunits[9–13].

The CBX2/4/6–8 subunits share an N-terminal chromodomain (CD) and a C-terminal Polycomb repressor (PcR) box domain[14]. The CD is a well-characterized methyl–lysine reader module that promotes CBX protein binding to methylated lysines in the histone H3 tail[15]. The PcR box mediates a direct binding of the CBX proteins to RING1A/B (and thereby to PRC1)[15]. Incorporating distinct CBX subunits into the PRC1 complexes provides structural diversity that facilitates diverse functional roles, as we and others have shown for a few cases[13, 16]. For instance, we have previously shown that although pluripotent embryonic stem cells (ESCs) express both the CBX6 and CBX7 subunits, the PRC1 complex predominantly contains the CBX7 subunit (together with PHC1, PCGF2, and RING1B). Further, in ESCs, PRC1-CBX7 represses CBX2, CBX4, and CBX8. During differentiation, CBX7 is downregulated, concomitantly with an upregulation of Cbx2 and Cbx4, which are incorporated into newly assembled

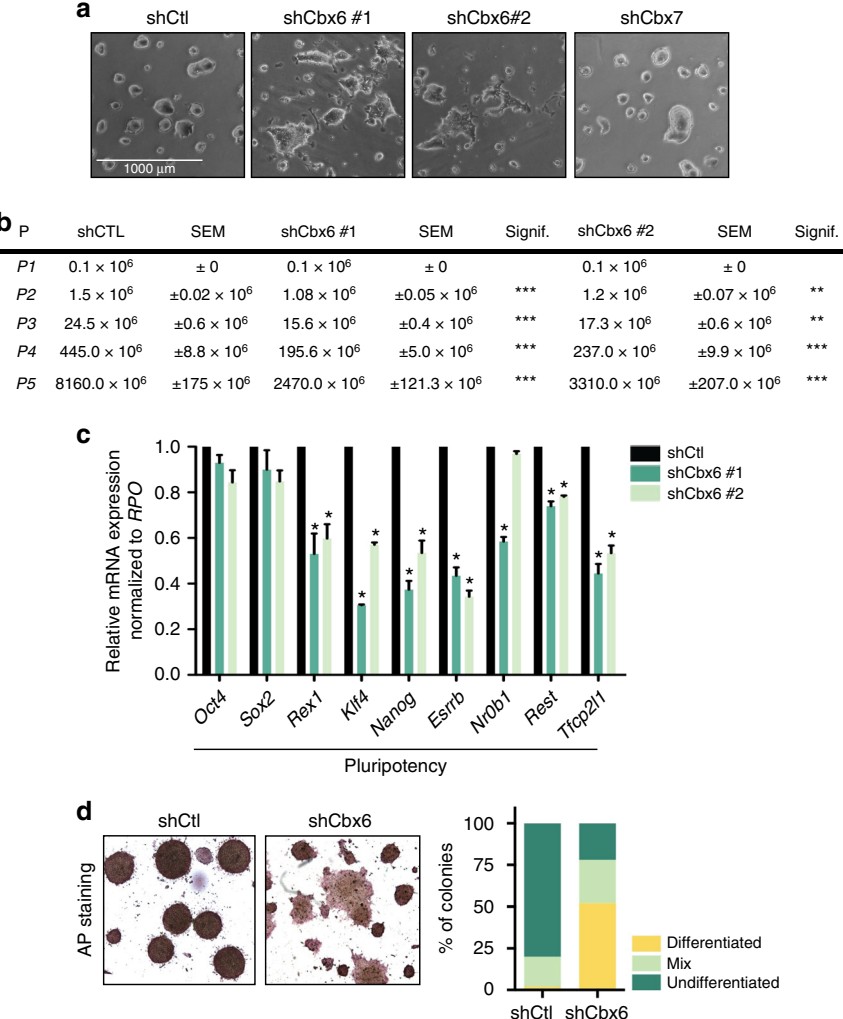

**Fig. 1** CBX6-depleted ESCs spontaneously differentiate. **a** Phase contrast image of shCtl, shCbx6, and shCbx7 ESC lines. **b** Accumulative proliferation measurements of shCtl and shCbx6 along passages. SEM, standard error of mean; P, passage; Signif., significance. Significance was analyzed through Student's t test. Significance was considered when P value was ≤0,05. P value of ** is ≤$10^{-2}$, P value of *** is ≤$10^{-3}$. **c** RT-qPCR analysis of control and CBX6-depleted ESCs. Results are shown relative to shCtl and are normalized to the housekeeping gene Rpo. Error bars represent standard deviation (SD) of three independent experiments. Significance was analyzed through Student's t test. Significance was considered when P value was ≤0.05 (*). **d** Phase contrast image of AP staining performed on shCtl or shCbx6 ESCs (right panel); quantification of the AP staining assays, representing the mean of three independent experiments in which around 40 random colonies were counted (left panel)

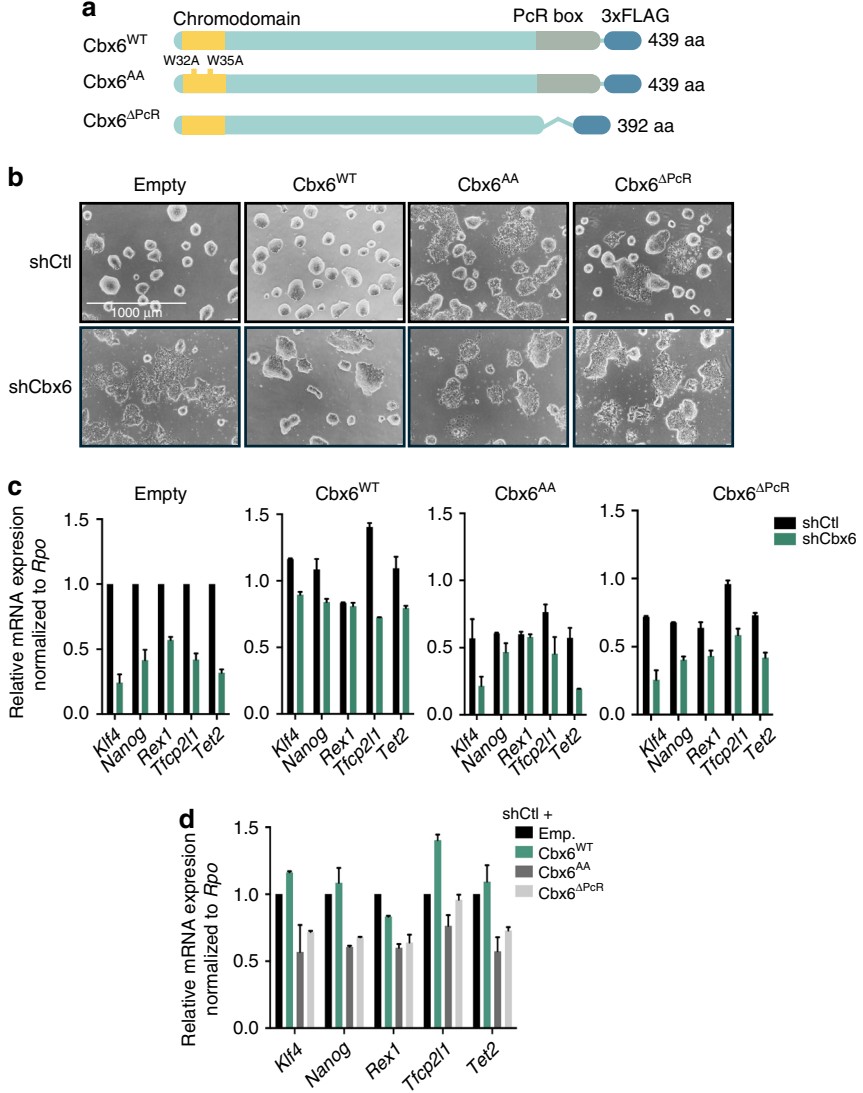

**Fig. 2** CBX6 function depends on its N- and C-terminal domains. **a** Schematic representation of the CBX6 constructs used in the rescue experiment. Note that every construct contained a silent mutation (not depicted) that conferred resistance to the CBX6 shRNA. **b** Phase contrast images of different cell lines overexpressing an empty construct or a CBX6^WT, CBX6^AA, or CBX6^ΔPcR construct, additionally infected with shCtl or shCbx6 lentiviral particles. **c** qRT-PCR analysis of pluripotency genes in the different cell lines. Results are shown relative to empty shCtl and were normalized to *Rpo*. Error bars represent SD of two independent experiments. **d** qRT-PCR analysis of pluripotency genes in the shCtl-infected cell lines. Results are shown relative to empty shCtl and normalized to *Rpo*. Error bars represent SD of two independent experiments

PRC1 complexes. Strikingly, CBX6 expression remains unaffected during differentiation, and its role in ESC pluripotency and embryonic development has remained elusive[13].

Here, we demonstrate that CBX6 is a key chromatin-associated factor required for balancing ESC pluripotency and differentiation. We find that CBX6 depletion induces rapid, spontaneous ESC differentiation. In contrast to the current paradigm, we show that CBX6 at the molecular level is present within both cPRC1 and ncPRC1 subunits. Notably, the CBX6 genome-wide distribution completely overlaps with the cPRC1 complex. Overall, our results indicate the presence of a PRC1 complex containing CBX6 in ESCs, which has a strong influence over self-renewal and pluripotency maintenance.

## Results

**CBX6 is required to maintain ESC identity.** To determine whether CBX6 is implicated in regulating ESC identity, we first

knocked down CBX6 using specific short hairpin RNAs (shRNA). ESCs were infected with lentiviruses harboring an shRNA control (shRNA-Ctl) or one of two independent shRNAs targeting *Cbx6* (sh*Cbx6* #1 and sh*Cbx6* #2). Both of these efficiently reduced CBX6 at the protein level (Supplementary Fig. 1a). We observed that CBX6 depletion consistently induced spontaneous differentiation, evidenced by flatter cell colony morphology and by the presence of fibroblast-like ESCs surrounding CBX6-depleted colonies (Fig. 1a). Consistent with previous reports, CBX7 depletion did not induce spontaneous differentiation (Fig. 1a)[13], suggesting that these two paralogue proteins have non-redundant functions.

We next investigated whether CBX6 is required for ESC self-renewal. Although CBX6-depleted ESCs could be sustained in culture for numerous passages, we recovered less cells than in the control condition (Fig. 1b), suggesting that self-renewal is compromised by the absence of CBX6. We verified that depletion of CBX6 was maintained over cell passages (Supplementary

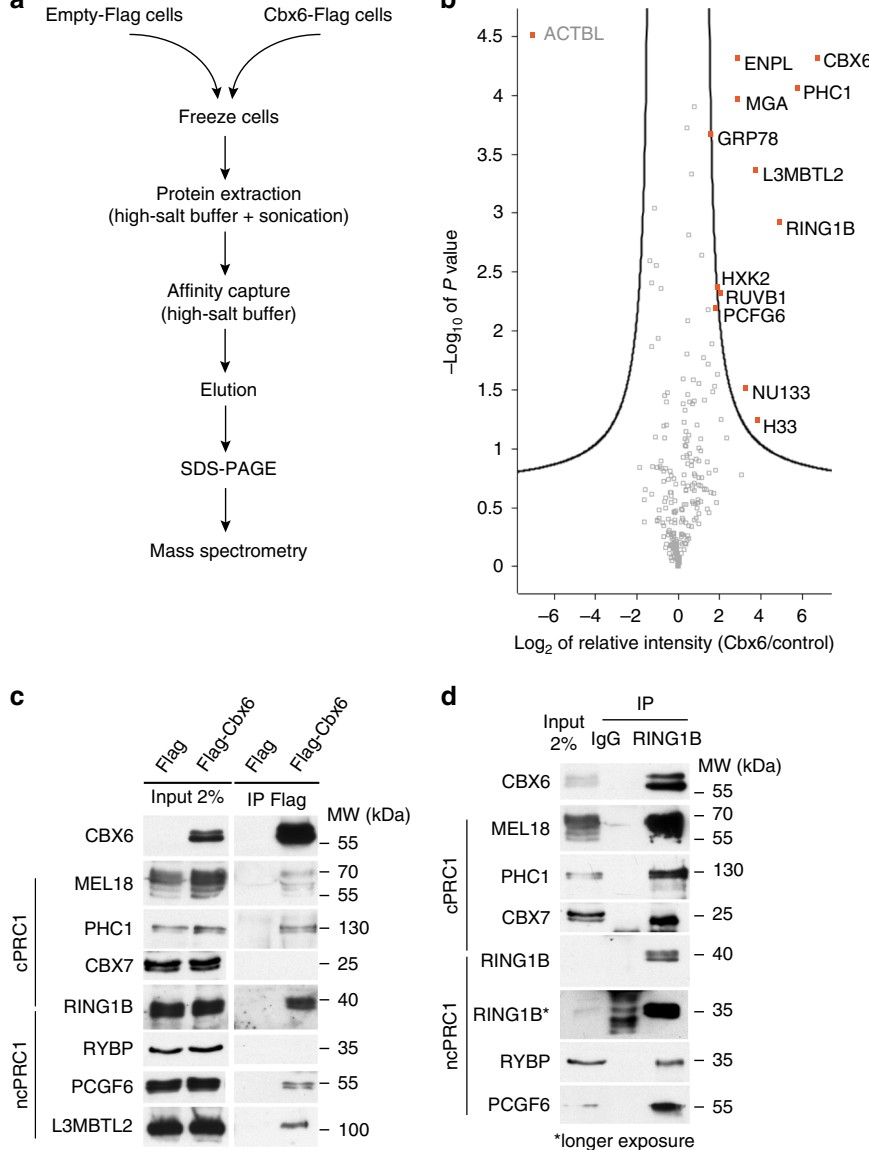

**Fig. 3** CBX6 interactome. **a** Protein complex affinity capture workflow. **b** Statistically enriched proteins in the 3×FLAG IP identified by permutation-based FDR-corrected *t*-test. The plot shows log$_2$ (difference) ratios of averaged protein intensities of the 3×FLAG pull-down over the control, plotted against the −log$_{10}$ (*P* value). The hyperbolic significance curve was calculated based on a combination of *P* value and fold-change. The proteins in the upper right corner represent the bait and its interactors (marked in red). A fold-change of ≥2 and a false discovery rate (FDR) of 0.05 were considered significant. **c** Co-IPs from total ESC extracts using an antibody against 3×FLAG. Western blots of different proteins are shown. **d** Co-IPs from total CBX6-3×FLAG-expressing ESC extracts using an antibody against RING1B. Western blots of different proteins are shown

Fig. 1b). Despite this observation, cell cycle progression was not affected upon CBX6 depletion (Supplementary Fig. 1c).

Consistent with the cellular phenotype, we detected a substantial downregulation of key pluripotency genes, such as *Rex1*, *Klf4*, *Nanog*, and *Esrrb*, upon *Cbx6* depletion (Fig. 1c). Moreover, alkaline phosphatase (AP) staining revealed that CBX6-depleted cultures contained a considerably higher percentage of colonies poorly stained by AP than control conditions (Fig. 1d). Indeed, strong AP staining was observed for more than 75% of the control colonies but for less than 25% of the CBX6-depleted colonies, showing a clear loss of AP staining in the absence of CBX6. Interestingly, CBX6 depletion did not result in spontaneous differentiation in 2i-cultured ESCs (Supplementary Fig. 1d). These results suggest that CBX6 might be dispensable in 2i-grown ESCs, or that its depletion is counterbalanced by the

addition of the two inhibitors, which safeguard ESCs from differentiation stimuli.

Thus, in contrast to the CBX7, the Polycomb subunit CBX6 has a unique function in ESCs, which contributes to the maintenance of ESC identity.

**The chromodomain and PcR box are essential for CBX6 function.** In order to gain insights into the mechanisms by which CBX6 sustains ESC identity, we engineered ESCs with a knock-down of endogenous *Cbx6* gene expression and that ectopically expressed either the wild-type or one of the CBX6 mutants. Two mutations were made within the two known functional domains present in all Pc-related proteins: in the first mutant, the 47 amino acids of the PCR box were removed (Cbx6$^{\Delta PcR}$), and in

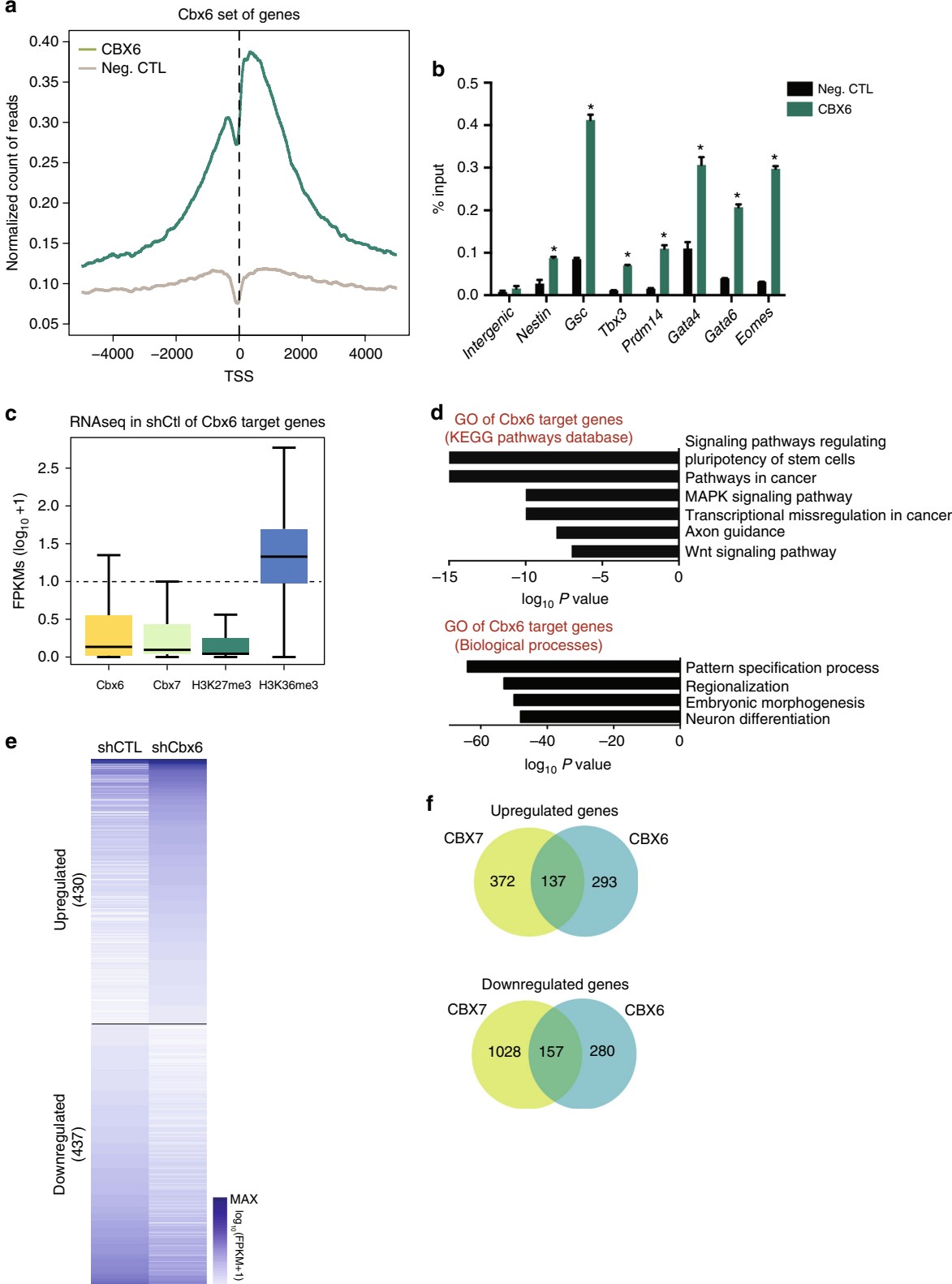

**Fig. 4** CBX6 genome-wide distribution features. **a** TSS (±5 kb) enrichment plot of CBX6 ChIP-seq at 2730 CBX6 target sites. **b** ChIP-qPCR validation of target genes of CBX6 in control and CBX6-3×HA-expressing cells. Results are shown relative to the input percentage. Error bars represent SD of three biological replicates. Significance was analyzed through Student's $t$ test. Significance was considered when $P$ value was ≤0.05 (*). **c** Box plots showing expression of CBX6, H3K27me3, and H3K36me3 target genes. **d** GO analysis of biological functions and signaling pathways of CBX6 target genes. $P$ values are plotted in −log. **e** RNAseq heat map of up- and downregulated genes in CBX6-depleted cells as compared to control cells. Only genes up- or downregulated by at least 1.5-fold as compared to control cells are shown. **f** Venn diagrams showing the overlap between CBX6 and CBX7 deregulated genes upon their depletion

the other, the two tryptophan residues of the aromatic cage of the chromodomain, which are essential for its proper folding and function, were substituted for glycine (Cbx6$^{AA}$)[15, 17] (Fig. 2a).

Only endogenous *Cbx6* mRNA was efficiently downregulated by CBX6-shRNA infection, and all three cell lines showed similar levels of the ectopically expressed proteins, as determined by western blot (Supplementary Fig. 2a, b). These cell lines allowed us to investigate whether the function of CBX6 in maintaining ESC identity was dependent on its interaction with RING1B and/or methylated lysines on histone H3.

As expected, CBX6$^{WT}$-expressing ESCs had the same round-shaped colony morphology as the control ESCs, and expression levels of several pluripotency genes were similar to those in control cells (Fig. 2b, c). In stark contrast, neither CBX6$^{\Delta PcR}$ nor CBX6$^{AA}$ could rescue the morphological phenotype (Fig. 2b) or sustain the expression of pluripotency genes (Fig. 2c).

Interestingly, the expression of CBX6$^{\Delta PcR}$ or CBX6$^{AA}$ proteins in shCtrl-infected cells resulted in notable cell differentiation, which was confirmed by AP staining, suggesting that CBX6$^{AA}$ and CBX6$^{\Delta PcR}$ mutants exert a dominant-negative effect over the endogenous CBX6 (Fig. 2d and Supplementary Fig. 2c). In summary, our data provide strong evidence that CBX6 functionality in ESCs relies on both the chromodomain and the conserved PcR box, linking CBX6 effects to chromatin binding and PcG interaction.

**CBX6 interacts with cPRC1 and ncPRC1 proteins**. To characterize the yet-unknown physical interactors of CBX6 in ESCs, we performed affinity capture of CBX6-3×FLAG combined with proteomic analysis by label-free, quantitative mass spectrometry (Fig. 3a). For this, we first generated an ESC line that stably expressed low levels of a Flag-tagged version of CBX6 (Supplementary Fig. 3a). CBX6-3×FLAG-expressing ESCs exhibited a normal morphology and expressed normal levels of the expected pluripotency markers as compared to parental E14 ESCs (Supplementary Fig. 3b, c), indicating that the ectopically expressed protein did not affect ESC identity. Importantly, CBX6 overexpression also did not affect the expression of PRC1 or PRC2, or the bulk H2AK119ub levels (Supplementary Fig. 3a). We then optimized the extraction and recovery of CBX6-3×FLAG from expressing cells (Supplementary Fig. 3a), using stringent conditions (with 400 mM NaCl and 0.5% Triton X-100). Pull-downs from cryomilled control and CBX6-3×FLAG ESCs were performed in three biological replicates. Only proteins identified in all replicas by mass spectrometry with a fold-change of ≥2 were considered as bona fide CBX6 interactors. We identified 11 significant interactors (Fig. 3b), including members of the cPRC1.2 complex (RING1B and PHC1). Unexpectedly, we also identified members of the ncPRC1.6 (or E2F6) complex (namely, L3MBTL2, MGA, and PCGF6)[10, 11]. These interacting partners were confirmed using specific antibodies (Fig. 3c) with CBX6-FLAG-expressing ESCs. The presence of CBX6 in PRC1 complexes was further confirmed by reverse immunoprecipitation experiments using a RING1B-specific antibody (Fig. 3d). We also observed an interaction between CBX6 and another Psc ortholog, PCGF2/MEL18, a member of the cPRC1 complex (Fig. 3c). In contrast, we did not detect CBX6 interactions with CBX7 or RYBP, neither by mass spectrometry nor by western blot (Fig. 3b, c), suggesting that CBX6-containing PRC1 complexes are distinctive from previously identified PRC1 complexes. It is important to note that CBX proteins and RYBP are mutually exclusive within the PRC1 complex context, as both compete for the same binding region of RING1B[8].

These data strongly support the existence of a CBX6-containing PRC1 complex(es) in ESCs, which is distinct from

CBX7- (Supplementary Fig. 3d, e) or RYBP-containing PRC1 complexes and has a critical role in supporting ESC identity.

**Genome-wide localization of CBX6 in ESCs**. We next aimed to identify the CBX6 target genes on the ESC genome. All commercially available antibodies we tested gave unsatisfactory or non-specific CBX6 binding, as revealed by their sustained peaks on CBX6-depleted ESCs. We therefore engineered an ESC line that expressed the endogenous CBX6 protein fused to a C-terminal triple HA tag, using CRISPR-Cas9. Importantly, the engineered CBX6-3×HA ESC cell line had a normal colony morphology, expressed normal levels of representative pluripotency and differentiation markers and was equivalent labeled with the stem cell CDy1 dye, as the parental ESCs (Supplementary Fig. 4a, b).

We next carried out chromatin immunoprecipitation (ChIP) followed by massive parallel sequencing (ChIP-seq) using anti-HA antibodies, in Cbx6-3×HA ESCs and parental ESCs (as a control). ChIP-seq analysis revealed a preferential distribution of CBX6 near transcription start sites (TSS) of genes (Fig. 4a and Supplementary Figs. 3c and 4b). We identified 16605 peaks of 2730 target genes (Fig. 4a and Supplementary Fig. 4b, c). Further validation by ChIP-qPCR experiments confirmed the presence of CBX6 in a subset of selected promoters (Fig. 4b).

CBX6 target genes were transcriptionally inactive or expressed at very low levels, comparable to the expression observed for CBX7 and H3K27me3-decorated genes (Fig. 4c). Gene ontology (GO) analysis indicated a significant enrichment of CBX6 on genes involved in developmental processes, such as regionalization and embryo morphogenesis (Fig. 4d). Also, genes occupied by CBX6 were implicated in signaling pathways regulating ESC pluripotency, such as the WNT and MAPK signaling pathways. Notably, these genes undergo activation upon differentiation of ESCs during embryo body formation, indicating that CBX6 is involved in governing developmental gene programs (Supplementary Fig. 4d).

In order to assess the impact of CBX6 on gene expression, we performed a genome-wide analysis of RNA levels by massive parallel sequencing on both control and CBX6-depleted cells. CBX6 depletion resulted in approximately equal numbers of upregulated (430) and downregulated (437) genes (Fig. 4e), which could indicate that CBX6 has a dual role in regulating gene transcription. Supporting this, gene expression and ChIP-seq data both revealed an overlap of 23 and 30% of up- and downregulated CBX6 direct target genes, respectively. Importantly, CBX6 loss led to downregulation of genes involved in development, including regulation of neurogenesis (Supplementary Fig. 4e). Upregulated genes were implicated in epithelium differentiation and tissue morphogenesis, possibly through the activation of signaling pathways involved in ESC biology, such as the MAPK signaling pathway (Supplementary Fig. 4e). Interestingly, CBX6 did not occupy any of the pluripotency genes downregulated after CBX6 depletion. This result indicated that the extinction of pluripotency was due to a secondary effect and provided additional support for an ESC differentiation model in which developmental gene programs co-regulate the extinction of the pluripotency network.

As mentioned, CBX7-depleted ESCs do not undergo differentiation. Thus, we analyzed changes in gene transcription observed upon CBX6 depletion and compared with those observed upon CBX7 depletion[13]. We found that 137 genes were commonly upregulated and 157 genes were commonly downregulated. In contrast, 293 and 372 genes were exclusively upregulated, while 280 and 1028 were exclusively downregulated in CBX6- and CBX7-depleted ESC, respectively (Fig. 4f). Further analysis of each of these gene sets did not highlight any

significant-enriched category that allowed us to explain CBX6- and CBX7-depletion phenotypic distinct consequences.

**CBX6 genome-wide distribution overlaps with PRC1.** We then tested to what extent, CBX6 target genes overlap with Polycomb complex target genes. Consistent with our MS data, we found that CBX6 target genes contained RING1B. Moreover, CBX6 targets were also decorated with PRC2 (as shown by the Suz12 subunit) as well as with CBX7 and H2AK119ub1 (Fig. 5a and Supplementary Fig. 5a). More specifically, 80.5, 85.8, 61.5, and 79.5% of

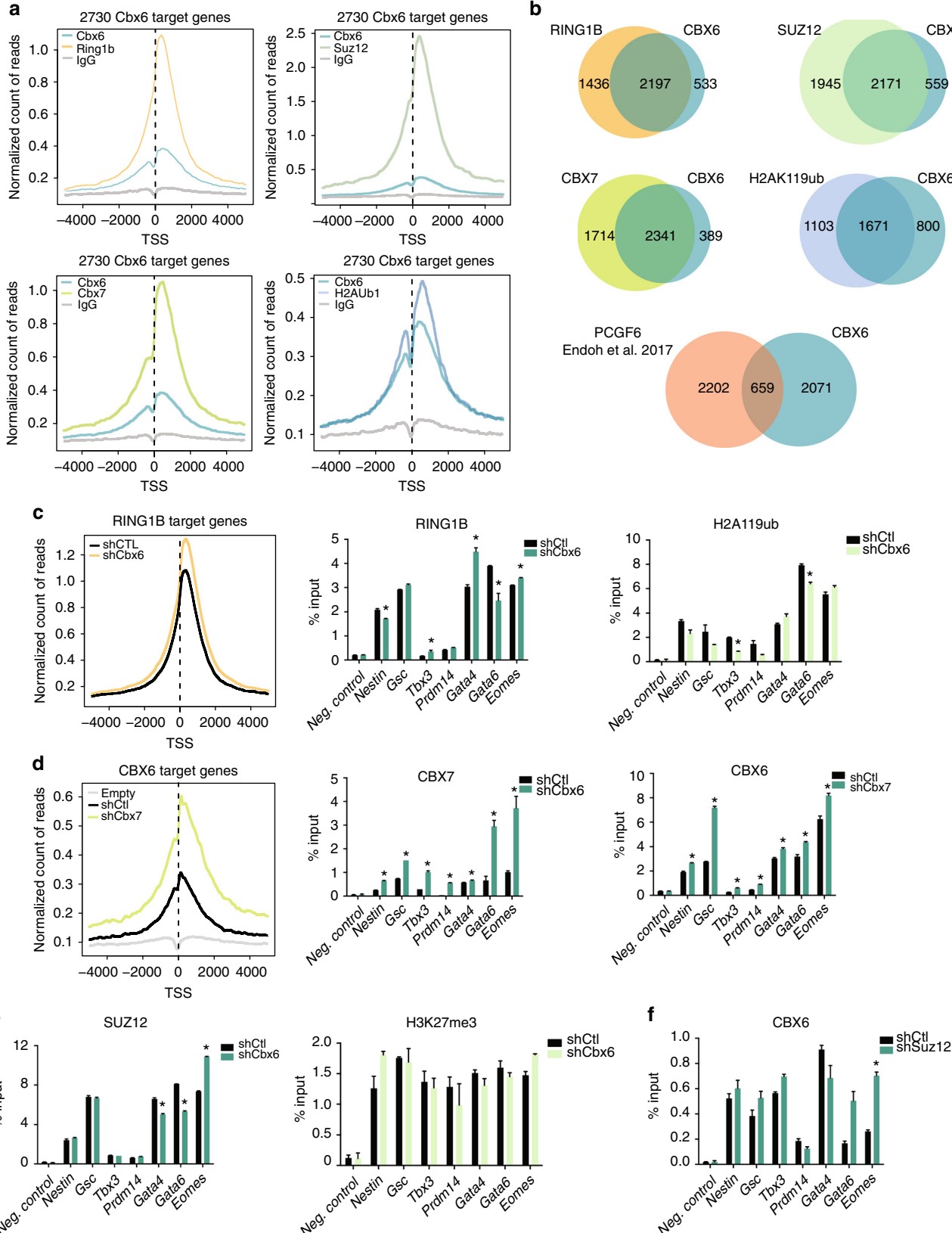

CBX6 target genes were co-occupied by RING1B, CBX7, H2AK119ub1, and Suz12, respectively (Fig. 5b). In contrast, CBX6 and PCGF6 shared limited number of target genes (24%) (Fig. 5b). Moreover, CBX6 genome-wide distribution was not affected upon PCGF6 depletion, suggesting that PCGF6 may be dispensable for CBX6 targeting to chromatin (Supplementary Fig. 5b). Therefore, a large set of CBX6 target genes are co-occupied by both cPRC1 and PRC2.

Since chromodomain-containing proteins are considered to be chromatin-targeting factors, we explored whether CBX6 contributes to targeting the cPRC1 complex. For this, we performed ChIP-qPCR experiments with RING1B in control or CBX6-depleted ESCs. Importantly, neither the protein levels of cPRC1 and PRC2 subunits nor the bulk level of selected histone modifications were affected by CBX6 depletion (Supplementary Fig. 5c, d). In stark contrast to CBX7 function[13], RING1B occupancy remained largely unaffected in CBX6-depleted ESCs (Fig. 5c). Although we noticed a general increase of RING1B level at its target genes, only 116 showed a significant gain of RING1B at their promoters, of which were 21 transcriptionally repressed. The catalytic activity of the cPRC1 complex was not affected, as H2AK119ub1 levels were not significantly perturbed following CBX6 knockdown (Fig. 5c). Nevertheless, we observed an increased CBX7 occupancy after CBX6 depletion (Fig. 5d). Interestingly, CBX7 depletion led to an upregulation of the CBX6 protein level (Supplementary Fig. 5e), followed by an increase of CBX6 occupancy at CBX6 target genes (Fig. 5d). These results suggest that neither CBX6 nor CBX7 can effectively compensate the loss of the other one. Further, neither Suz12 nor H3K27me3 distribution were majorly affected by CBX6 depletion (Fig. 5e). Therefore, major transcriptional changes observed upon CBX6 depletion cannot be explained by changes in cPRC1/PRC2 distribution and/or function.

Previous reports showed that CBX7, but not CBX6, binds to H3K27me3 in vitro[18]. We studied the effect of ablating PRC2 function on the chromatin recruitment of CBX6. We stably depleted Suz12 using shRNAs in both control ESCs and CBX6-3×HA ESCs (Supplementary Fig. 5f). In contrast to PRC2-dependent CBX7 binding (Supplementary Fig. 5g), we found that CBX6 binding was not PRC2-dependent, as CBX6 recruitment to sites lacking PRC2 was either unchanged or even slightly increased (Fig. 5f). Overall, these results suggest that CBX6 recruitment to chromatin was not dependent on H3K27me3, indicating that CBX6 and CBX7 in ESCs do not exert overlapping functions.

## Discussion

Our data show that CBX6 plays a role in maintaining ESC identity, by preserving the balance between the states of pluripotency and differentiation. CBX6 depletion resulted in spontaneous differentiation of ESCs, indicating that CBX6 regulates the expression of gene networks that control these fundamental biological functions. In CBX6-depleted ESCs, expression of genes regulating the central pluripotency network (i.e., Klf4, Nanog, and Rex1) was strongly downregulated. Concomitantly, there was a premature expression of differentiation markers (i.e., Brachyury, Pax6, and Otx2). Analysis of CBX6 direct target genes with deregulated expression revealed that both the WNT and the MAPK signaling pathways were perturbed in CBX6-depleted ESCs. Based on these findings, we speculate that an alteration in the expression of these pathways may result in the collapse of the balance between self-renewal and lineage commitment, ultimately resulting in differentiation.

Notably, depletion of different PRC1 complex members results in very specific phenotypes. For instance, and in contrast to CBX6, CBX7 is not required to preserve ESC self-renewal, yet its depletion greatly affects PRC1 and PRC2 recruitment to chromatin[13]. Similarly, RING1B-mutant ESCs can be sustained in culture even though expression of RING1B targets are strongly affected[19, 20]. However, complete impairment of PRC1 function (following a Ring1A/B double depletion) strongly reduces the capacity for self-renewal[21]. The finding that CBX6-depleted ESCs exhibited a phenotype not observed after depletion of a single PRC1 component suggests that CBX6 has a PRC1-independent—or at least a cPRC1-independent—role, and that other family members cannot compensate for its deletion. We believe that the phenotypic differences observed upon CBX6 or CBX7 depletion could be due to a specific combination of gene expression changes that would trigger differentiation upon CBX6 depletion, but not upon CBX7 depletion. Some of the most interesting differentially expressed genes include Nanog (exclusively downregulated in shCbx6 cells) or Cbx6 itself (upregulated in shCbx7 cells). In addition, a large number of genes involved in extrinsic signaling pathways (such as Wnt and MAPK) were also deregulated. Fine-tuning the balance between multiple and opposing signals downstream of these pathways generates contrasting functional outcomes, either maintaining self-renewal (i.e., in CBX7-depleted ESC) or instructing lineage differentiation (i.e., in CBX6-depleted ESC).

Because of its homology with CBX2, −4, −7, and −8[14], CBX6 had been proposed to form part of the cPRC1 complex, although there was little scientific evidence for this. Specifically, an interaction between CBX6 and RING1B and PHC2 was observed in HeLa cells[22], and more recently, an interaction between CBX6 and the cPRC1 complex was observed in neural stem cells[23]. Nevertheless, these interactions have never been demonstrated in the context of ESCs, where CBX7 is the main CBX protein incorporated into the cPRC1 complex[13, 23]. We have now revealed a physical interaction between CBX6 and members of both cPRC1 and ncPRC1, such as MGA, PCGF6, and L3MBTL2.

In line with these results, the genome-wide distribution of endogenous CBX6 strongly overlapped with that of cPRC1, supporting the validity of our MS data. The CBX6 protein levels are very low in ESCs, which is likely why CBX6 has not previously been observed in PRC1 purifications from ESCs[10, 24]; we believe that overexpression was essential for enabling the co-capture of CBX6 partners. Our results may reflect the existence of a single unique complex or, alternatively, of several distinct CBX6-containing complexes. Therefore, the next critical step will be to determine the exact composition of the CBX6 complex(es).

---

**Fig. 5** CBX6 occupies cPRC1 sites. **a** TSS (±5 kb) enrichment plot of CBX6, RING1B, CBX7, H2AK119ub, and SUZ12 ChIP-seq at 2730 CBX6 target sites. **b** Venn diagrams showing the overlap of CBX6 target genes with those of RING1B CBX7, H2AK119ub[13], SUZ12[13], and PCGF6[35]. **c** (left) TSS (±5 kb) enrichment plot of RING1B in shCtl and shCbx6 ESCs, at 2730 CBX6 target sites. (right) ChIP-qPCR in shCTL or shCBX6 ESCs of RING1B and H2AK119Ub. **d** (left) TSS (±5 kb) enrichment plot of CBX6 in shCtl and shCbx7 ESCs, at 2730 CBX6 target sites. (right) ChIP-qPCR in shCTL and shCBX6 (or shCbx7) ESCs of CBX7 (or CBX6), respectively. **e** ChIP-qPCR in shCTL or shCBX6 ESCs of Suz12 and H3K27me3. **f** ChIP-qPCR of CBX6 in shCTL and shSuz12 ESCs. **c–f** For all the experiments, an intergenic region was used as a negative control gene. Results are shown relative to percentage of input. For all the experiments an intergenic region was used as a negative control gene. Results are shown relative to percentage of input. Error bars represent SD of three biological replicates. Error bars represent SD of three biological replicates

Although the genomic landscape of CBX6 is almost identical to that of cPRC1, its depletion in ESCs does not compromise the binding of cPRC1 or PRC2 core subunits, suggesting that most of the cPRC1 recruitment relies on CBX7. Interestingly, we observed a slightly increased binding of CBX7, but not of H2AK119ub, upon CBX6 depletion. Similarly, CBX7 deletion had a partial effect on CBX6 binding to chromatin, by modestly increasing it. Thus, we hypothesize that within the context of the cPRC1 complex both CBX6 and CBX7 possess partially overlapping functions. It is likely that cPRC1–CBX7 acts as the main complex, whereas cPRC1–CBX6 complex has a minor role in gene repression and functions as a redundant failsafe for CBX7 functions. However, cPRC1–CBX6 acquires specific role during ESC differentiation, when cPRC1–CBX7 complex has been dismantled. This potential role requires further investigation.

Intriguingly, and in contrast to CBX7[13], we found that CBX6 localization to chromatin was not impaired in ESCs depleted of functional PRC2, indicating that CBX6 is not recruited to its target genes via H3K27me3. Unlike *Drosophila* Pc, which tightly binds to H3K27me3, the mammalian counterparts display different affinities for different histone modifications. For instance, peptide pulldown experiments revealed that the CBX6 chromodomain does not recognize H3K27me3[18]. Although the aromatic cage pocket of the chromodomain is essentially indistinguishable between CBX paralogues, the amino acids outside but bordering the aromatic cage have subtle variations, with an adjacent hydrophobic pocket that is required for stabilizing the interaction between the CBX proteins and the methyl residues. Milosevich and coworkers showed that the CBX6 hydrophobic pocket cannot bind the conserved histone alanine at the −2 position of the tri-methyl lysine site H3K27me3 (ARKS motif)[25, 26]; instead, a hydrophobic residue in the position −2 of the methyl lysine is key for CBX6 binding. This suggests that CBX6 might associate with another, yet-unidentified histone modification.

## Methods

**ESC culture and embryoid body differentiation**. Wild-type (E14Tg2A) ESCs were cultured feeder free in plates coated with 0.1% gelatin. Coating was achieved by covering the plates with gelatin 15 min at 37 °C. After removing any remaining gelatin, ESCs were cultured with Glasgow minimum essential medium (Sigma) supplemented with β-mercaptoethanol, sodium pyruvate, penicillin–streptomycin, non-essential amino acids, GlutaMAX, 20% fetal bovine serum (Hyclone), and leukemia inhibitory factor (LIF).

**AP staining**. ESCs ($1 \times 10^3$) were cultured in a six-well plate for 5 days. The AP assay was performed with the alkaline phosphatase detection kit (Millipore) following the manufacturer's instructions.

**Calcium phosphate transfection**. HEK-293T cells ($2 \times 10^6$) were plated onto a p10 plate. The following day, the calcium phosphate-DNA precipitates were prepared by pooling together the plasmid in 0.25 M $CaCl_2$. While vortexing, calcium phosphate-DNA solution was added dropwise to an equal volume of HBS 2× (HEPES-buffered saline solution, pH 7.05, of 0.28 NaCl, 0.05 M HEPES, and 1.5 mM $Na_2HPO_4$) at room temperature. After 15 min at room temperature, the solution was added to the HEK-293T cells for lentivirus production.

**Lentivirus production and infection**. Lentivirus was produced by transfecting HEK-293T packaging cells with 5 μg of pCMV-VSV-G, 6 μg of pCMVDR-8.91, and 7 μg of the pLKO-shRNA (Sigma) plasmid (either pLKO-shCTL or pLKO-shCBX6), using the calcium phosphate transfection method. Cells were incubated with the transfection mix for no more than 16 h, after that the medium was replaced by ESC (LIF-free) fresh medium. After 48 h, lentiviral particles were collected and filtered using a 0.45 μm filter. For infection, $2 \times 10^5$ target cells were plated in a six-well plate. Two ml of the filtered medium containing the lentiviral particles was used to culture the cells in the presence of LIF (1:500) and Polybrene (1:1000) was added to the culture medium. The following morning medium was replaced with fresh medium for selection. Infected cells were selected using the appropriate antibiotic (2 μg ml$^{-1}$ of puromycin, 50 μg ml$^{-1}$ of hygromycin, or 50 μg ml$^{-1}$ of G418) for 3 days.

**Protein extract preparation and western blot analysis**. Whole cell extracts for western blot analysis were prepared in lysis buffer IP300 (50 mM Tris-HCl pH 7.6, 300 mM NaCl, 10% glycerol, and 0.2% NP-40). Lysates were incubated for 5 min on ice and then sonicated five cycles (30 s ON/30 s OFF) in a Bioruptor (Diagenode). Cell extracts were centrifuged for 25 min at maximum speed at 4 °C. Protein concentration was quantified by Bradford assay (Bio-Rad) according to the manufacturer's instructions. Samples were analyzed by SDS-PAGE using acrylamide gels in running buffer (25 mM Tris-base, 200 mM glycine, 0.1% w/v SDS) at 100 V. Proteins were transferred onto nitrocellulose membranes at 300 mA for 70 min at 4 °C in transfer buffer (25 mM Tris-HCl, pH 8.3, 200 mM glycine, 20% v/v methanol). Protein transfer was checked by Ponceau S (Sigma) staining. Transferred membranes were blocked with 5% w/v milk in TBS-Tween (10 mM Tris-HCl, pH 7.5, 100 mM NaCl and 0.1% Tween-20) for 30 min with rotation at room temperature. Blocked membranes were incubated overnight with the primary antibody (with 5% w/v milk in TBS-Tween) at 4 °C with rotation. The next day, membranes were washed twice for 5 min with TBS-Tween followed by incubation with the secondary antibody conjugated to the horseradish peroxidase (1:5000, Dako), diluted in TBS-Tween, for 1 h at room temperature. After two 15-min washes with TBS-Tween at room temperature, proteins were detected by an enhanced chemiluminiscence reagent (Pierce ECL Western Blotting Substrate, Thermo Scientific). All primary antibodies, and the conditions for their use are listed in Supplementary Information as Supplementary Table 1.

**RNA extraction and cDNA synthesis and gene expression analysis**. RNA was extracted with the RNeasy mini kit (Qiagen) following the manufacturer's instruction. cDNA was synthesized by reverse transcription using a cDNA synthesis kit (Quanta-Bioscience). Real-time PCR reactions were performed using SYBR Green I PCR Master Mix (Roche) and the Roche LightCycler 480. Expression was normalized to the housekeeping gene *Rpo*. Expression was normalized to the housekeeping gene *Rpo*. All the primers used are listed in Supplementary Information as Supplementary Table 2.

Paired-end RNA-sequencing was performed using 1 μg RNA and two samples per lane to achieve maximum sequencing depth. The genomics unit performed the quality control and library preparation. The libraries were sequenced using Illumina HiSeq2000 sequencer. Genes with a fold-change of 1.5 were considered to be differentially expressed.

**Chromatin immunoprecipitation**. Two 15-cm plates for each cell line to be tested were prepared at 70–80% confluency. Cells were trypsinized and crosslinked in 1% formaldehyde for 10 min at room temperature in a shaker. To stop the fixation reaction, 0.125 M glycine was added to the existing culture media and incubated for 5 min. Sample pellets were then washed twice with PBS 1× at room temperature. After aspirating PBS completely, crosslinked pellets were resuspended in 1.3 ml ice-cold IP buffer (1× volume SDS buffer [100 mM NaCl, 50 mM Tris-HCl, pH 8., 5 mM EDTA, pH 8, and 2% SDS] and 0.5 volume Triton dilution buffer (100 mM NaCl, 50 mM Tris-HCl, pH 8.6, 5 mM EDTA, pH 8 and 5% Triton X-100]) with proteinase inhibitors. Samples were sonicated for 12 min (30 s ON/30 s OFF) in a Bioruptor (Diagenode) at maximum output. After sonication, samples were centrifuged at 4 °C at maximum speed for 20 min. To check chromatin size, 20 μl of the supernatant was mixed with 80 μl of 1× PBS and de-crosslinked for 3 h at 65 °C in a shaker (1000 rpm), followed by PCR purification kit (Qiagen). DNA was eluted in 30 μl of water and quantified by nanodrop. Around 800 ng was loaded in a 1% agarose gel. If chromatin was between 200 and 500 bp, 40 μg of chromatin was used to ChIP proteins. ChIP reactions consisted of 40 μg of chromatin and 5 μg of antibody to a final volume of 500 μl. ChIP reactions were incubated overnight at 4 °C on rotation. The next day, 30 μl of washed agarose beads were added to the ChIP reactions and incubated for 2 h at 4 C. After incubation, beads were washed three times with 1 ml of low-salt buffer (140 mM NaCl, 50 mM HEPES, pH 7.5, and 1% Triton X-100) and once with 1 ml high-salt buffer (500 mM NaCl, 50 mM HEPES, pH 7.5, and 1% Triton X-100). All supernatant was removed, and 110 μl of freshly prepared elution buffer (1% SDS, 100 mM NaHCO3) was added per ChIP reaction (including 1% of every input). Samples were de-crosslinked at 65 °C for 3 h in a shaker (1000 rpm). DNA was purified following the PCR purification kit. DNA was eluted with 100 μl of water (with two consecutive elutions of 50 μl). A sample of 2 μl was used for ChIP-qPCR analysis.

For histone modifications, 5 μg of chromatin was used.

ChIP experiments using the CBX6-3×HA cell line were performed with the ChIP-IT High Sensitivity Kit from Active Motif (53040) according to the manufacturer's instructions.

**Rescue experiment**. The CBX6 cDNA resistant to shRNA#52 (CBX6R) was produced by inserting a silent mutation in the 10th nucleotide position of the shRNA#52 recognition site using the QuickChange Site-Directed Mutagenesis kit (Stratagene), following the manufacturer's instructions. The CBX6-FLAG mutant for the chromodomain (CBX6$^{AA}$) was generated by mutating CBX6R at the tryptophans in positions 33 and 36, switching them to glycine. The CBX6-FLAG mutant for the PcR box (CBX6 $^{\Delta PcR}$) was generated by depleting the PcR box domain.

For rescue experiments, the CBX6-FLAG versions (CBX6R, CBX6$^{AA}$, and CBX6$^{\Delta PcR}$) were first overexpressed, and the cell lines were infected with pLKO-shCTL or pLKO-shCBX6 once they were established.

**Proliferation curve**. About 100,000 cells for each condition were plated in a six-well plate. Every 2 days, cells were trypsinized, and 100,000 cells were re-plated, for five passages.

**CRISPR-Cas9 vector construction**. sgRNAs (5′-TTTCTTGGCTTTATA-TATCTTGTGGAAAGGACGAAACACC-3′, 5′-GACTAGCCTTATTT-TAACTTGCTATTTCTAGCTCTAAAAC-3′) targeting the CBX6 locus were designed using the online software: http://crispr.mit.edu. Primers coding for the sgRNAs were annealed and assembled with a pX459 (Puromycin selection) and a pX458 (GFP selection) vectors (Addgene) using the method described by Zhang et al.[27]. Briefly, vectors were digested using the *Bbs*I restriction enzyme 30 min at 37 C. In parallel, sgRNAs were phosphorylated using T4 PNK (NEB) and annealed in the thermocycler using the following parameters: 30 min at 37 C; 5 min at 95 C, and then ramp down to 25 C at 5 C min$^{-1}$. Gel-purified digested vectors were ligated with phosphorylated sgRNAs. Ligation reaction was incubated at 10 min at room temperature. Targeting efficiency was calculated using the T7 endonuclease surveyor assay.

**Donor vector construction**. Left and right CBX6 homology arms were generated by PCR using specific primers and cloned by Gibson Technology into the HDR donor vector. To create the 3× HA tag construct, two oligonucleotides were generated to amplify the tag, which also contained an overlapping sequence specific to the vector, in order to insert the fragment into the HDR vector using a Gibson reaction.

**Stable cell line generation**. sgRNAs cloned into the different vectors (pX458 and pX459, 6 µg) and the HDR linearized vector (3 µg) were co-transfected in $3 \times 10^6$ mES cells by nucleofection (Nucleofection Amaxa kit) and incubated for 24 h. Cells transfected with the pX459 vector were selected for 48 h with puromycin (1 µg ml$^{-1}$); cells transfected with pX458 were checked for GFP efficiency using a fluorescence microscope.

After single-cell sorting by FACS using GFP fluorescence or size, cells were plated into two 96-well plates (96 clones per condition). After growth, genomic DNA was extracted from each clone and analyzed by PCR to check if the tag had been successfully inserted. Positive clones were sequenced.

**Immunoprecipitation followed by western blot analysis**. ESCs expressing FLAG-tagged constructs were lysed in IP300 buffer (50 mM Tris-HCl, pH 7.6, 300 mM NaCl, 10% glycerol, 0.2% NP-40) supplemented with protease and phosphatase inhibitors. Cells were sonicated five cycles (30 s ON/30 s OFF) on a Bioruptor sonicator (Diagenode) followed by full-speed centrifugation. An aliquot of protein (1 mg) was incubated with 30 µl of prewashed FLAG M2 beads (Sigma) (with IP300) and incubated usually for 1 h (depending on the IP) on a rotating wheel at 4 °C. Samples were washed three times with IP300 buffer. Elution was performed by incubating the dried beads with 60 µl of 2× Laemli buffer (Roth Karlsruhe) at 100 °C, or with 0.2 µg ml$^{-1}$ FLAG peptide in PBS, as appropriate, for 15 min. For endogenous IPs, 1 mg of protein was incubated with the antibody for 1 h, followed by incubation of A or G sepharose beads for 2 h at 4 °C. Elution was performed with 2× Laemli buffer (Roth Karlsruhe) at 100 °C for 15 min. Uncropped scans of Fig. 3 western blots are provided as Supplementary Fig. 6.

**CBX6-3×FLAG affinity capture followed by mass spectrometry analysis**. Affinity capture was carried out on cryomilled material from empty control cells or CBX6-3×FLAG cells as described previously[28, 29]. Briefly, cells cultured as described above were collected by scraping, washed in PBS, and the resulting wet cell pellet was dripped into liquid nitrogen producing frozen pellets. The pellets were then subjected to cryomilling in a Retsch PM-100: three milling cycles of 3 min each (reverse rotation, 1 min interval, no break time) at 400 rpm. The milling jar was cooled with liquid nitrogen between cycles. The resulting powder was collected and stored at −80 °C. Cell powder (900 mg) was used for protein extraction in each replicate; experiments were performed in triplicate for both sample and control. Extraction was performed with 20 mM HEPES, pH 7.4, 400 mM NaCl, and 0.5% v/v Triton X-100. Samples were briefly sonicated to improve protein extraction from chromatin (25 × 2 s, 1 Amp pulses, with 1 s pause between pulses using a QSonica S4717 microtip probe). After sonication, samples were centrifuged at 20,000 RCF (relative centrifugal force) at 4 °C for 10 min. Supernatants were combined with 27 µl of anti-FLAG beads slurry for 30 min at 4 °C. Beads were eluted with 20 µl of 1× LDS at 70 °C for 5 min with mixing. The eluates were loaded onto NuPAGE 4-12% Bis-Tris gels (Invitrogen) and run until the dye front migrated ~6 mm into the gel. The gel was stained with Coomassie Brilliant Blue G-250, the samples were excised (gel plugs), and then further processed, as described below.

**Sample preparation for mass spectrometry and data analysis**. Gel plugs were processed essentially as previously described[28]. Briefly, prior to loading on the gel, the samples were alkylated with iodoacetamide; after electrophoresis, the gel plugs were cut into ~1 mm cubes for processing. Samples were destained with several washes of 50% v/v acetonitrile (ACN) in 50 mM ammonium bicarbonate at 37 °C with shaking. Destained gel pieces were dehydrated by washing with 100 µl ACN, and placed in a speed-vac for ~10 min at RT. Trypsin working solution was added and gel pieces were allowed to swell on ice and then were incubated at 37 °C to undergo tryptic proteolysis. Trifluoroacetic acid (TFA) was added to each tube (2% w/v final concentration), and incubated 5 min at RT. The supernatant was recovered and transferred to a 0.5 ml microfuge tube (tryptic digest supernatant). An aliquot of 50 µl 0.1% w/v TFA was added to the gel pieces, which were extracted a further 45 min at RT with shaking. The supernatants were removed and pooled with the appropriate tryptic digest supernatant. Pooled extracted peptides were desalted using reversed-phase OMIX tips (Agilent P/N A57003100) as per the manufacturer's instructions. The peptides were eluted from the tips first with 100 µl of aqueous 40% (v/v) ACN, 0.5% (v/v) acetic acid (E1) and then with 100 µl of 80% (v/v) ACN, 0.5% (v/v) acetic acid (E2). E1 and E2 were combined, frozen in liquid nitrogen, and dried in a centrifugal vacuum concentrator.

For MS analysis, dried peptide samples were resuspended in 10 µl of aqueous 5% (v/v) methanol, 0.2% (v/v) formic acid. Mass spectra were recorded on a Orbitrap Fusion mass spectrometer (Thermo Fisher Scientific).

Database searching and label-free quantitation were performed by MaxQuant 1.5.2.8 using the UP000000589 mouse database[30]. The match between run feature was disabled, and intensities were based on maximum peak height. The "proteingroups.txt" file was uploaded to Perseus 1.5.3.0, and protein identifications from the decoy database were removed. LFQ intensities were logarithmized. Control experiments were grouped together, as were CBX6-3×FLAG pull-down experiments. Proteins were filtered, with the constraint that at least one group (CBX6 or control) should contain at least three valid values. Missing values were imputed from a normal distribution. A two-sample Student's *t* test was performed with an arbitrary fold-change of ≥2 required for significance and a permutation-based FDR = 0.05 used for truncation.

**Bioinformatic analysis**. The ChIP-seq samples were mapped against the mm9 mouse genome assembly using Bowtie with the option –m 1 to discard those reads that could not be uniquely mapped to just one region[31]. MACS (Zhang et al., 2008) was run with the default parameters but adjusting the shiftsize to 75 bp to perform the peak calling, and each set of target genes was retrieved by matching those ChIP-seq peaks in the region from 2.5 Kb upstream of the TSS to the transcriptional end site as annotated in RefSeq.

The plots showing the distribution of ChIP-seq reads 5 Kb around the TSS of each target gene set were generated by counting the number of reads in this region for each gene (according to RefSeq), and then averaging this value with the total number of mapped reads of each sample and the number of targets of the gene set.

The heat maps displaying the density of ChIP-seq reads 5 kb around the TSS of each target gene set were generated by counting the number of reads on this region for each individual gene and normalizing this value with the total number of mapped reads of the sample. Genes on each ChIP heat map are ranked by the logarithm of the average number of reads on the same genomic region.

GO and other term enrichment analyses were done using Enrichr web-based tools[32].

The RNAseq samples were mapped against the mm9 mouse genome assembly using TopHat[33] with the option –g 1 to discard those reads that could not be uniquely mapped to just one region. Cufflinks[34] was run to quantify the expression in FPKMs of each annotated transcript in RefSeq. Genes showing one or more FPKMs are considered to be expressed. Up- and downregulated gene lists in control compared to knockdown samples were generated by applying a fold-change threshold of 1.5.

**Data availability**. All the ChIP-seq and RNAseq raw and processed files generated in this manuscript have been deposited in the NCBI GEO under the accession number GEO: GSE98723. The mass spectrometry proteomics data have been deposited to the ProteomeXchange Consortium with the dataset identifier PXD007577.

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

## Acknowledgements

We thank Dr Pasini for providing a-PCGF6 antibodies. We thank the CRG Genomics, and Biomolecular Screening and Protein Technologies Units. We also thank all members of Di Croce's laboratory for discussion, and V.A. Raker for help in preparing the manuscript. This work was supported by grants from the Spanish Ministry of Economy and Competitiveness (BFU2016-75008-P), Centro de Excelencia Severo Ochoa 2013-2017 (SEV-2012-0208), AGAUR, Fundació "La Marató de TV3," CERCA Programme/Generalitat de Catalunya, and EU FP7 Programs 4DCellFate (277899) to L.D.C. This work aided by collaboration with the National Center for Dynamic Interactome Research and the National Resource for the Mass Spectrometric Analysis of Biological Macromolecules, supported in part by NIH grant P41 GM109824 to Michael P. Rout and grant P41 GM103314 to Brian T. Chait. This paper is subject to the NIH Public Access Policy.

## Author contributions

A.S., L.M. and L.D.C. designed the study; A.S. conducted all the experiments, except the 3xFLAG-Cbx6 pulldown and mass spectrometry analysis, carried out by J.L., H.J. and K.R.M. and the validation of the interactome, conducted by M.S.; E.B. performed the bioinformatics analysis; L.M. and L.D.C. supervised the experiments and provided intellectual support toward design and interpretation of the results; A.S. and L.D.C. wrote the manuscript.

## Additional information

**Competing interests:** The authors declare no competing financial interests.

