## [Peer Review File · Nature Communications]

Reviewers' comments:

Reviewer #1 (Remarks to the Author):

Santanach et al., characterize in their manuscript entitled "The polycomb group protein CBX is an essential regulator of embryonic stem cell identity" the role of the so far understudied CBX protein CBX6 in ESC pluripotency. The manuscript contains very interesting data (e.g. the finding that CBX6 is member of canonical and non-canonical PRC complexes as well as effects of its depletion on pluripotency), is well written and clearly advances the field. It should be published in Nature Communications, however a few issues should be addressed

1. The CBX6 complex purification was performed in an overexpression system. This should be discussed in the manuscript. In Figure 3d it is not clear if it is endogenous CBX or the overexpressed Flag fusions. Why is there no Ring1B in the input in Figure 3d ?
2. How does CBX6 depletion affect the cell cycle ? The authors should address this.
3. The authors finish the manuscript with a statement that CBX6 might associate with another histone modification. Here it might be interesting to speculate which one it might be. Could it be RNA binding ?
4. Can the authors comment clearer on the overlap of CBX6 genome-wide distribution with canonical and non-canonical PRC1 ? There is a better overlap with cPRC1 members than PCGF6. What can be the reason for this ? What is the overlap between PCGF6 and other cPRC1 members ?

Reviewer #2 (Remarks to the Author):

In the manuscript "The Polycomb group protein CBX6 is an essential regulator of embryonic stem cell identity", Santanach and collaborators reported the characterization of Cbx6 function(s) in mouse embryonic stem (ES) cells. The data presented indicate that Cbx6 is part of both canonical and non-canonical PRC1 complexes. Unexpectedly, depletion of CBX6 triggers ES cell differentiation, which is different than the effect of Cbx7 depletion. The authors also provided an extensive characterization of the interplay between PRC1 and PRC2 with CBX6.

Overall, this is an interesting and nicely performed study, which highlights a novel role for Cbx6 in mouse ES cells. The manuscript is well-written, and its concise message will be of interest of experts within the field of both gene regulation and stem cell.

I have no major criticism, but I am including some comments below, that might help in improving the impact of this manuscript.

1. It is unclear if Cbx6 is required in naive ground state. To address this the authors should investigate the function of Cbx6 in 2i-grown ES cells.
2. From the data presented in figure 1, it seems that it seems that Cbx6 depletion might also affect cell cycle. The authors should explore this possibility.
3. I am aware that obtaining a good quality data for Pcgf6 ChIPseq has been proved to be difficult so far. Yet, I encourage the authors to attempt such approach.

Reviewer #3 (Remarks to the Author):

This is a very interesting manuscript describing an essential role for Cbx6 in the maintenance of embryonic stem (ES) cells. The most exciting finding of this study is that Cbx6 knockdown did not substantially affect target occupancy by Ring1B, in contrast to knockdown of Cbx7, which recruits canonical PRC1 (cPRC1)-Ring1B to its targets through binding to H3K27me3. Indeed, Suz12 knockdown did not reduce the occupancy of Cbx6 at its typical targets, indicating that H3K27me3 is not necessary for Cbx6 recruitment to these loci. This is surprising, especially given that Cbx6 largely co-occupies genomic loci with Ring1B, Suz12, and H2Aub1 and appears to require its chromodomain (which may mediate association with methylated lysines) and its PcR box domain (which may mediate association with PcG components) for support of ES cell maintenance. The authors suggest that Cbx6 maintains ES cells through functions independent of PRC2 and cPRC1-Ring1B, although the mechanism remains unclear.

Santanach et al. also identified some PRC1.6 components (L3MBTL2, MGA, PCGF6), in addition to PRC1.1 components, as interactors of Cbx6. They did not investigate the significance of these interactions in this manuscript, however.

Overall, this paper has unveiled an unexpected role for Cbx6 in ES cells that will be of great interest to readers in the field of regulation of stem cell fate through chromatin modification. However, I feel that the manuscript lacks some crucial information regarding the nature of this role of Cbx6.

Major points

1. The novel and interesting aspects of this study rely on comparison between Cbx6 and its conventional counterpart, Cbx7. The authors should therefore compare the results for Cbx6 and Cbx7 side by side in some experiments such as the identification of interactors (Fig. 3) and the modulation of gene expression patterns (Fig. 4e, Supplemental Fig. 4).
2. The mechanism by which Cbx6 regulates the maintenance of ES cells remains elusive, even though this is the central part of the study. Some data related to this point should be presented. For example, are the PRC1.6 components necessary for the role of Cbx6 in ES cells? It would be desirable to show the genome-wide effects of Cbx6 knockdown on Ring1B and H2Aub1 occupancy as well as on PRC1.6 components (such as L3MBTL2, which also plays an essential role in ES cell maintenance [Qin et al. Cell Stem Cell 2012]). Although the authors state that "RING1B occupancy remained largely unaffected in CBX6-depleted ES cells," it is possible that Cbx6 regulates a small subset of Ring1B targets via PRC1.6 and that these targets are responsible for Cbx6 function in supporting ES cell maintenance.

Minor points

1. P values should be added for the results in bar graphs.
2. The quality of immunoblots in Supplementary Fig. 5 (especially b and e) is poor. For example, it is difficult to believe that the level of H3K27me3 is unchanged in Cbx6-depleted cells. It should be confirmed that the signals are not saturated, and signal intensities should be quantified.

Response to reviewer #1

1. The CBX6 complex purification was performed in an overexpression system. This should be discussed in the manuscript. In Figure 3d it is not clear if it is endogenous CBX or the overexpressed FLAG fusions. Why is there no Ring1B in the input in Figure 3d?

As requested, we have now specified throughout the text and figures when working in overexpression conditions.

We agree with this reviewer that the input of Ring1B is difficult to observe, although it is visible in the original X-ray film. As the IP has been performed using the Ring1B antibody which is extremely efficient in immunoprecipitating Ring1B, we cannot see the input within the same exposure time that was used to monitor the immunoprecipitated material. We now include in the manuscript a longer exposure time in which the input is perceptible.

Co-IP of RING1B using α FLAG antibody

2. How does CBX6 depletion affect the cell cycle? The authors should address this.

We thank the referee for raising this important point. We have performed a BrdU proliferation assay in control and CBX6-depleted ESC to analyse cell cycle progression. Our results show that CBX6 depletion does not affect cell cycle progression. We have now included this information as part of Supplementary figure 1 in the new version of the manuscript.

Analysis of cell cycle progression in control and CBX6-depleted ESC

3. The authors finish the manuscript with a statement that CBX6 might associate with another histone modification. Here it might be interesting to speculate which one it might be. Could it be RNA binding?

We thank the referee for raising this point. In order to identify which histone modification is recognized by CBX6, we have performed several experiments including histone peptide arrays and peptide pull-down assays. However, we did not get satisfactory results (as described below) and we finally decided not including this information in the manuscript. The difficulty on getting this information could be attributed to many reasons. CBX6-GST recombinant protein production is extremely challenging, and the resulting purified protein is easily degraded, which could interfere in its proper binding to the histone peptides. Moreover, CBX6-GST protein precipitates if frozen and must be freshly generated for every test, introducing variability to the experiments. Finally, it should be also noted that we used an *in vitro* recombinant assay devoid of any other factor that may be required for methyl-histone recognition. As suggested by the reviewer, CBX6 histone recognition could depend on other factors such as RNA for an optimal histone binding. It has been previously reported that CBX6 displays affinity to bind to RNA (Bernstein et al. 2006 Mol Cell Bio) although we did not explore this part.

As mentioned in the manuscript, Milosevich and co-workers showed that CBX6 hydrophobic pocket cannot bind the conserved histone alanine at the -2 position of the trimethyllysine site H3K27me3 (ARKS); rather, a hydrophobic residue in the position -2 of the methyllysine is key for CBX6 chromodomain. In our histone peptide array and histone peptide pulldowns we observed that CBX6 could have a bias to preferably bind histone H4. We propose either H4R23 (RKVLRDNIQ) or H4K31 (QGITKPAIR) as possible CBX6-binding sites. Using the H4R23 binding site would require an arginine residue to occupy the aromatic cage of CBX6. This may sound unusual for a chromodomain, but is not unprecedented for other reader-protein families. Using the H4K31 site would require an isoleucine residue to be tolerated at the -2 position. Methylation at both the H4R23 and H4K31 sites have been identified in proteomic PTM surveys (PhosphositePlus), but these have not been further studied to date. It would be very interesting in the future to investigate the possible binding of CBX6 to these modifications.

4. Can the authors comment clearer on the overlap of CBX6 genome-wide distribution with canonical and non-canonical PRC1? There is a better overlap with cPRC1 members than PCGF6. What can be the reason for this? What is the overlap between PCGF6 and other cPRC1 members?

We thank the referee for this comment. In Figure 5b we show that CBX6 is mostly overlapping with canonical PRC1 subunits, rather than with the non-canonical PRC1 subunit tested, PCGF6. A possible explanation for this could be that CBX6 predominantly acts within the canonical PRC1 complex, and this is somehow reflected in its genome-wide distribution. However, we also need to take into account that there could be a technical reason behind this observation, as the PCGF6 ChIP-seq used for this analysis (Endoh. et al 2017) was performed in ESC (R1) with a different genetic background than our ESC (E14). Moreover, we need to consider

that the use of different ChIP protocols as well as differences in the sequencing conditions may also affect the output of the ChIP-seq signal. Because of these reasons, we decided to perform our own PCGF6 ChIPseq experiment using a commercially available antibody from LScBio and using the same experimental conditions as the rest of our experiments were performed. We obtained a smaller number of target genes (418), from which around 60% overlapped with the published data set (Endoh. et al 2017).

Venn diagram showing the overlap between the published data set and our generated data.

Qualitatively PCGF6 ChIPseq data from Endoh et al., contained higher peaks and less background, whereas the background level in our ChIPseq profile was higher.

At first sight, a substantial fraction of the peaks reported by Endoh and colleagues is not identified in our PCGF6 ChIPseq, because the background signal is higher. Below we show a region containing promoters targeted by PCGF6.

UCSC screenshot example.

Further analysis of the TSS peak distribution revealed that our ChIPseq was indeed capturing some of the best peaks of the published Endoh experiment. Below, we show the ChIPseq signal strength of the three gene sets (genes detected in both experiments, genes detected in one of them). On the left, we see in grey that the 258 genes identified in both ChIPseqs are the ones with the highest signal, while the set of 2603 genes reported by Endoh correspond to weaker peaks. On the right we confirm the same results in our ChIPseqs.

Graphical distribution of normalized count of reads 5 kb upstream and downstream of TSS.

Overall, we consider that both the reliability and the quality of the LScBio PCGF6 ChIPseq experiment are not optimal, and that is why we believe it is much more appropriate to work with a set of genes that has already been published.

As requested by this referee, we tested the overlap between PCGF6 and other cPRC1 members such as CBX7 and MEL18 (PCGF2). We could observe that the degree of overlap between PCGF6 and these two proteins was similar compared to what was observed between PCGF6 and CBX6. In particular, 23%, 15.2% and 32.5% of PCGF6 target genes overlapped with CBX6, CBX7 and MEL18 target genes, respectively.

Venn diagrams showing the overlapping of different PRC1 subunits.

From the analysis of these new comparisons, we could confirm that the overlapping between PCGF6 and canonical and non-canonical is quite similar. These results suggest that canonical and non-canonical share the same targets. We would like to stress that, apart from the role of Cbx6 in ESCs pluripotency and self-renewal, the main novelty of our manuscript is that we biochemically demonstrated that Cbx6 is part of the non-canonical PRC1.

Looking at the multiple Venn diagram it is difficult to understand within which complex (canonical or non-canonical) every subunit tested is acting. In fact, mutually exclusive proteins, such as CBX and PCGF family members, share the same target genes.

Response to Reviewer #2

1. It is unclear if *Cbx6* is required in naïve ground state. To address this the authors should investigate the function of *Cbx6* in 2i-grown ES cells.

We thank the referee for raising this important point. To address it, we stably knocked down CBX6 in 2i-cultured ES cells. Interestingly, CBX6 depletion did not result in spontaneous differentiation as documented in the bright field image (A), neither by quantification of the Alkaline Phosphatase staining (B). At the molecular level, we monitored the expression of representative pluripotency and differentiation genes. Our data indicate that their expression was not affected (C). These results suggest that CBX6 might be dispensable in 2i-grown ESCs, or that its depletion is counterbalanced by the addition of the 2 inhibitors (Chiron and PD03), which safeguard ESCs from differentiation stimuli (such as the absence of CBX6 function). We have now included these new data in the revised version of our manuscript (Supplementary figure 1d).

Cbx6 depletion in 2i cultured ESC

2. From the data presented in figure 1, it seems that *Cbx6* depletion might also affect cell cycle. The authors should explore this possibility.

We agree with this referee that this is an important point. We have performed a BrdU proliferation assay in control and CBX6-depleted ESC and analysed cell cycle progression. Our results show that CBX6 depletion does not affect cell cycle progression, despite differentiation. We have now included this information as part of Supplementary figure 1 in the revised version of our the manuscript.

Analysis of cell cycle progression in control and CBX6-depleted ESC

3. I am aware that obtaining a good quality data for Pcgf6 ChIPseq has been proved to be difficult so far. Yet, I encourage the authors to attempt such approach.

We carried out a ChIP-seq for PCGF6 using an available commercial antibody (LScBio) under the same experimental conditions as the other endogenous ChIPs in the manuscript were performed. ChIP-seq analysis revealed a lower number of target genes (418), from which around 60% overlapped with the published data set (Endoh. et al 2017).

Venn diagram showing the overlap between the published data set and our generated data.

Qualitatively PCGF6 ChIPseq data from Endoh et al., contained higher peaks and less background, whereas the background level in our ChIPseq profile was higher that it was difficult to call the peaks, which is why many were not detected.

At first sight, a substantial fraction of the peaks reported by Endoh and colleagues is not identified in our PCGF6 ChIPseq, because the background signal is higher. Below we show a region containing promoters targeted by PCGF6.

UCSC screenshot example.

Further analysis of the TSS peak distribution revealed that our ChIPseq was indeed capturing some of the best peaks of the published Endoh experiment. Below, we show the ChIPseq signal strength of the three gene sets (genes detected in both experiments, genes detected in one of them). On the left, we see in grey that the 258 genes identified in both ChIPseqs are the ones with the highest signal, while the set of 2603 genes reported by Endoh correspond to weaker peaks. On the right we confirm the same results in our ChIPseqs.

Graphical distribution of normalized count of reads 5 kb upstream and downstream of TSS.

Overall, we consider that both the reliability and the quality of the LScBio PCGF6 ChIPseq experiment are not optimal, and that is why we believe it is much more appropriate to work with a set of genes that has already been published.

Response to Reviewer #3

Major points:

1. The novel and interesting aspects of this study rely on comparison between Cbx6 and its conventional counterpart, Cbx7. The authors should therefore compare the results for Cbx6 and Cbx7 side by side in some experiments such as the identification of interactors (Fig. 3) and the modulation of gene expression patterns (Fig. 4e, Supplemental Fig. 4).

We thank the reviewer for this comment. CBX7 interactome has been extensively characterized in ESC (Tavares et al. 2012, Cell). In these cells, CBX7 mainly interacts with members of the canonical PRC1 complex. We have added a comparative table listing CBX6 and CBX7 interactors as part of Supplementary Figure 3.

CBX6 interactors	CBX7 interactors
RING1B	RING1B
PCGF2 (MEL18)	PCGF2 (MEL18)
PHC1	PHC1
PCGF6 (MBLR)	PCGF6 (MBLR)
L3MBTL2	
MGA	

In addition, to corroborate those interactions, we have performed several new immunoprecipitation experiments for CBX7 using the same conditions as for CBX6 (Figure 3c). Our data confirm that CBX7 is not interacting with the non-canonical subunits L3MBTL2 and RYBP. Importantly, it interacts with PHC1, PCGF2 and RING1B and PCGF6, as previously described. These results have now been included as Supplementary Figure 3e.

We have also carefully analysed the changes in gene transcription observed upon CBX6 or CBX7 depletion. We have overlapped up- and down-regulated genes from CBX6- and CBX7-depleted ESC: 137 and 157 genes were up- and down-regulated in common, respectively. In contrast 293 and 372 genes were exclusively up-regulated, while 280 and 1028 were exclusively down-regulated in CBX6- and CBX7-depleted ESC, respectively.

Venn diagram showing the overlap between different CBX6 and CBX7-depleted ESC

We further analysed the GO terms of each gene list (CBX6-specific, common, or CBX7-specific for up and downregulated genes) using the Enrichr software (<http://amp.pharm.mssm.edu/Enrichr/>), however, we did not find any significant enriched category that allowed us to explain the dissimilar phenotypic features between both cell-lines.

We believe that the phenotypic differences observed upon CBX6 or CBX7 depletion could be due to a specific combination of gene expression changes that would trigger differentiation in the case of CBX6 depletion, but not upon CBX7 depletion. Some of the most interesting differentially expressed genes include Nanog (exclusively downregulated in shCbx6 cells) or Cbx6 itself (upregulated in shCbx7 cells). In addition, we found a large number of genes involved in extrinsic signaling pathways (such as Wnt and MAPK), which are deregulated. Fine-tuning the balance between multiple and opposing signals downstream of these pathways generates contrasting functional outcomes, either maintaining self-renewal (*i.e.* in CBX7-depleted ESC) or instructing lineage differentiation (*i.e.* in CBX6-depleted ESC).

These new interesting data are now included in the revised version of our manuscript as part of Figure 4.

2. The mechanism by which Cbx6 regulates the maintenance of ES cells remains elusive, even though this is the central part of the study. Some data related to this point should be presented. For example, are the PRC1.6 components necessary for the role of Cbx6 in ES cells?

We thank the referee for this comment. To address this point, we have now stably depleted PCGF6 in ESC, using an shRNA construct that efficiently reduced PCGF6 at protein level. We then performed ChIPseq of CBX6 in shCTL and shPCGF6 ESC. Our new data indicate that CBX6 genome-wide distribution is not affected upon PCGF6 depletion, suggesting that PCGF6 may be dispensable for CBX6 targeting at chromatin. This is now included in the reviewed version of our manuscript as Supplementary Fig. 5b.

It would be desirable to show the genome-wide effects of Cbx6 knockdown on Ring1B and H2Aub1 occupancy as well as on PRC1.6 components (such as L3mbtl2, which also plays an essential role in ES cell maintenance [Qin et al. Cell Stem Cell 2012]).

As requested, we performed ChIP-seq for RING1B in shCtrl and shCbx6. We obtained 3633 RING1B target genes in shCtrl ESC, and 3814 in shCbx6 ESC, 3,411 of which were common to both. Graphical distribution of normalized count reads around the TSS showed an increase of RING1B occupancy in CBX6-depleted cells. Below we include a screenshot representing this.

Ring1 occupancy in control and Cbx6-depleted ESC

To elaborate a more quantitative approach of these observations, we measured the ChIP signal levels in the promoters of the full set of genes in the genome to identify directly those presenting a significant gain (or loss) after CBX6 depletion. In particular, we selected those genes in which the ChIPseq level 500 bp around the TSS of one sample over the other one was above 1.75-fold change (FC). We identified 116 genes that significantly exhibited higher RING1B ChIP-seq levels, and 351 genes that lost RING1B. As genes losing RING1B already had very low levels of RING1B in their promoters, we considered them to be non-specific; we thus focused on the cohort of genes that gained RING1B.

We next overlapped the 116-gene list with the RNAseq data and found that 21 genes that gained RING1B in their promoters were downregulated following CBX6 depletion. Overall, CBX6 depletion did affect Ring1B distribution on a specific sub-set of gene. Of those, this gain of RING1B resulted in a decrease of transcription for 21 genes. Thus, the reduction in transcription of the rest of downregulated genes (416) is likely due to an indirect mechanism.

We also carried out a ChIP-seq for PCGF6 using an available commercial antibody (LScBio) using the same experimental conditions as the other endogenous ChIPs in the manuscript were performed. ChIP-seq analysis revealed a lower number of target genes (418), from which around 60% overlapped with the published data set (Endoh et al 2017).

Venn diagram showing the overlap between the published data set and our generated data.

Qualitatively PCGF6 ChIPseq data from Endoh et al., contained higher peaks and less background, whereas the background level in our ChIPseq profile was higher that it was difficult to call the peaks, which is why many were not detected.

At first sight, a substantial fraction of the peaks reported by Endoh and colleagues is not identified in our PCGF6 ChIPseq, because the background signal is higher. Below we show a region containing promoters targeted by PCGF6.

Further analysis of the TSS peak distribution revealed that our ChIPseq was indeed capturing some of the best peaks of the published Endoh experiment. Below, we show the ChIPseq signal strength of the three gene sets (genes detected in both experiments, genes detected in one of them). On the left, we see in grey that the 258 genes identified in both ChIPseqs are the ones with the highest signal, while the set of 2603 genes reported by Endoh correspond to weaker peaks. On the right we confirm the same results in our ChIPseqs.

Graphical distribution of normalized count of reads 5 kb upstream and downstream of TSS.

Overall, we consider that both the reliability and the quality of the LScBio PCGF6 ChIPseq experiment are not optimal, and that is why we believe it is much more appropriate to work with a set of genes that has already been published.

Although the authors state that “RING1B occupancy remained largely unaffected in CBX6-depleted ES cells”, it is possible that Cbx6 regulates a small subset of Ring1B targets via PRC1.6 and that these targets are responsible for Cbx6 function in supporting ES cell maintenance.

Venn diagram showing the overlap between different ChIPseq data sets

To address the referee's suggestion, we have overlapped CBX6, CBX7, PCGF6, RING1B and MEL18 ChIP-seq data to find a possible subset of target genes specific for CBX6, RING1B and PCGF6. As shown in the figure, there are only 5 genes targeted by CBX6, RING1B and PCGF6, seeming unlikely the hypothesis that these 3 factors regulating these genes could be responsible for Cbx6 function.

Minor points:

1. P values should be added for the results in bar graphs.

As suggested, P values have been included in the experiments where the number of replicates was 3 or more.

2. The quality of immunoblots in Supplementary Fig. 5 (especially b and e) is poor. For example, it is difficult to believe that the level of H3K27me3 is unchanged in Cbx6-depleted cells. It should be confirmed that the signals are not saturated, and signal intensities should be quantified.

As suggested by this referee, we have now improved the quality of immunoblots in Supplementary Figure 5.

For Supplementary Figure 5d non-saturated images have been obtained, and signal intensities have been quantified. Now we can clearly state that CBX6 depletion does not affect the bulk levels of these histones.

For Supplementary Figure 5f, we have obtained higher quality images for CBX7 and H3K27me3 western blots.

REVIEWERS' COMMENTS:

Reviewer #1 (Remarks to the Author):

The authors addressed the concerns raised and the paper should be published in Nature Com.

Reviewer #2 (Remarks to the Author):

In the revised version of the manuscript, Santanach and colleagues implemented all the suggestions from this and the other reviewers. The manuscript now contains new interesting experiments and analyses that definitely reinforce and support the authors' conclusions. I have no further comments.

Reviewer #3 (Remarks to the Author):

The authors have adequately addressed the issues I raised.